# Four Trials Is Not Enough: The Amount of Prior Audio–Visual Exposure Determines the Strength of Audio–Tactile Crossmodal Correspondence Early in Development

**DOI:** 10.3390/bs15091184

**Published:** 2025-08-30

**Authors:** Shibo Cao, Rong Tan, Vivian M. Ciaramitaro

**Affiliations:** 1Department of Psychology, Developmental and Brain Sciences, University of Massachusetts Boston, Boston, MA 02125, USA; shibo.cao001@umb.edu; 2Department of Counseling Psychology, University of Massachusetts Boston, Boston, MA 02125, USA

**Keywords:** Bouba–Kiki, sound–shape correspondence, development, crossmodal, visual experience, touch

## Abstract

Successfully navigating the world involves integrating sensory inputs and selecting appropriate motor actions. Yet, what information belongs together? In addition to spatial and temporal factors, correspondence across sensory features also matters. In the Bouba–Kiki (BK) effect, spiky shapes are associated with sounds like “kiki”, and round shapes are associated with “bouba”. Such associations exist between auditory and visual (AV) and auditory and tactile (AT) stimuli, where objects are only explored via touch. Visual experience influences AT associations, which are weak in early blind adults and in fully sighted 6- to 8-year-olds, who have a more naïve visual experience. It has been found that prior AV exposure in children enhances AT associations. Here, we consider how the amount of prior AV exposure strengthens AT associations. Sixty-one 6- to 8-year-olds completed four or eight AV trials, which involved seeing a round and spiky shape and indicating which shape best matched a sound. Then, children completed 16 AT trials: feeling a round and spiky shape. Shapes were hidden from view, and children had to indicate which of the two shapes best matched a sound. We found that eight, but not four, trials of *prior AV exposure* enhanced AT associations. Our findings suggest that the amount, not just the type, of prior exposure is important in the development of audio–tactile associations.

## 1. Introduction

Our sensory systems are constantly bombarded by multiple sources of information. This presents an interesting challenge for our brains, determining which signals arising from different senses—multisensory signals—originate from the same object in the world. Multisensory signals are more likely to arise from a common source and be integrated if they are either spatially or temporally congruent, meaning they appear from the same spatial location, or change in the same way over time (i.e., [53]; [49]; [52]; [12]). For example, we associate the sight of a dog and the sound of its barking since both the visual and auditory signals arise from the same location in space and are synchronized in time. Besides spatial and temporal factors, we are also more likely to perceive select sensory information from our different senses as related, a preferential mapping of features across the senses, known as crossmodal correspondence (reviewed in [48]). For example, we tend to associate larger objects with low-frequency sounds and smaller objects with high-frequency sounds, as is the case for musical instruments, where a bass, a larger string instrument, has a lower-frequency voice than a violin, a smaller string instrument. One type of sound–shape correspondence that has received a fair amount of study is the tendency to map abstract rounded shapes with nonsense words such as “maluma” and abstract angular shapes with “takete” ([32]). This particular correspondence has come to be known as the Bouba–Kiki (BK) effect; Ramachandran and Hubbard first coined this term, since they used “bouba” and “kiki” as sounds in their experiments ([41]).

BK sound–shape correspondence has been found for individuals from different cultures ([18]) speaking different languages containing different speech sounds and orthographies ([17]; [11]; [6]; [30]), and even in remote cultures with no written language ([8]). Furthermore, associations have been found across different senses, such as between nonsense sounds and tastes that are sweet versus bitter, spicy versus mild, effervescent versus non-effervescent ([15]; [16]; [21]) or between nonsense sounds and objects that are touched and felt to be round versus spiky ([23]; [47]; [14]) or smooth versus rough ([20]).

One long-standing question, which is the focus of much research, is when BK associations first emerge in development. Some have argued that BK audio–visual associations between abstract shapes and nonsense sounds (AV) may subserve sound symbolism, the non-arbitrary mapping of language sounds and word meaning. The early emergence of BK AV associations in pre-linguistic infants would suggest an important role for these associations in language learning. According to [41] ([41]), the BK effect may stem from reinforced links between sensory and motor regions in the brain, influencing the trajectory of early language development ([41]). Although some studies have identified these sound–shape mappings in infants as early as 4 months ([38]) or 11 months of age ([1]), the evidence is inconsistent, with other studies documenting only weak BK associations ([39]; [45]). A meta-analysis across 11 studies in children aged 4 to 38 months reported small effect sizes and weak BK associations, which gradually became more pronounced during the first 3 years of life. The ability to map nonsense sounds to abstract shapes may follow, not precede, language acquisition, as 3-year-olds struggle to apply this correspondence to unfamiliar words, whereas 5- or 7-year-olds demonstrate this generalization ([55]). Extending this work to older children, Chow and Ciaramitaro found that 6- to 8-year-olds also exhibited weaker BK associations than older children or adults, with BK associations strengthening with age and with stronger associations for “bouba” over “kiki” pseudowords ([13]), replicating effects in younger children, who also show stronger associations for “bouba” over “kiki” pseudowords ([22]). Taken together, these results suggest that BK effects for AV sound–shape correspondence are still relatively weak early in development, even in children as old as 8 years of age.

While many studies have considered the development of sound–shape correspondence between heard nonsense words (e.g., “bouba” or “kiki”) and seen abstract shapes (e.g., round/spiky), for audio–visual associations (AV; [35]; [38]; [1]; [39]; [55]; [22]; [14]), less is known about the development of sound–shape correspondence between heard nonsense words and abstract shapes that are experienced via touch—audio–tactile associations (AT). As is the case for BK AV associations, BK AT associations are also weak early in development. Children 6 to 8 years of age do not exhibit AT associations significantly above chance; only 9- to 11-year-olds show AT associations above chance ([14]).

Both innate and learned mechanisms may be important in forming BK associations. What experiences strengthen the BK sound–shape correspondence over the course of development? Below, we focus on the role of experience in determining the strength of audio–tactile BK associations.

### 1.1. The Role of Visual Experience in Strengthening AT Associations

Previous work highlighted the important role of visual experience in audio–tactile sound–shape correspondence. Studies examining associations between 3-D tactile shapes with “bouba” versus “kiki” nonsense sounds found differences between typically sighted and blind adults, with age at which vision loss occurred being critical. While late-blind adults show the same AT associations as typically sighted adults, early blind adults show weakened or absent AT associations ([23]). Furthermore, being deprived of visual input by twelve years of age results in weakened associations, but blindness after the age of 12 does not have as great an effect ([47]), suggesting a sensitive period when visual input is particularly important.

It is not just congenitally blind or early blind adults who show weak AT associations; even typically sighted children show weak AT associations ([14]; Experiment 1). Although typically sighted children are not visually impaired, they have a more limited, or naïve, experience with visual shapes and objects in the world, especially for the abstract and less familiar shapes used in testing BK associations. Thus, one would predict that providing complementary visual exemplars of the abstract shapes that are later explored via touch might help strengthen subsequent AT associations in early childhood. As predicted, when typically sighted children were given prior exposure to the same BK sounds and 2-D renditions of the 3-D shapes they would later explore only via touch during 16 AV trials, AT associations were strengthened ([14]; Experiment 2). This prior AV experience enhanced children’s use of more effective and efficient manual strategies when subsequently exploring objects only via touch ([10]).

### 1.2. The Role of Practice in Strengthening AT Association

In addition to early visual experience influencing the strength of AT associations ([23]; [27]; [47]), time-on-task, or practice, has also been shown to matter. When adults manually explored a pair of round and spiky shapes, four trials of a raised outline of a form, and then four trials of the same raised form but filled in, and then judged which shape best matched a BK nonsense word, performance varied as a function of visual experience as well as time-on-task. While blind adults initially showed no AT associations on trial 1, AT associations emerged with repeated AT exposure of the same shapes, such that more participants matched the correct shape to the given nonsense word on trial 4, as well as on trials 5 or 8 when the different, related shape pair was introduced. Unlike blind individuals, sighted but blindfolded adults showed AT associations from the beginning, trial 1, but association strength diminished with repeated AT exposure ([26]). Overall, these effects of time-on-task suggest that AT associations were strengthened when additional AT exposure, four trials, provided extra information in the intact sensory system that participants tend to rely on. Thus, additional audio–tactile experience benefited blind adults who rely more on touch, their intact sensory modality, but not blindfolded typically sighted adults, who might rely more on visual cues for shape information. Of note, other studies in the blind have also found weak or absent AT associations ([23]; [47]), but most previous studies have not considered how association strength changes with time on task.

Typically sighted 6- to 8-year-olds also show weak AT associations. However, unlike blind adults, repeated AT exposure does not enhance AT association strength in 6- to 8-year-old children. [14] ([14]) asked 6- to 8-year-old children to manually explore a pair of round and spiky shapes, hidden from view, and judge which shape best matched a BK nonsense word for a total of 16 AT trials. Shapes were of uniform rigidity, texture, and thickness, only differing in shape contour. With repeated AT exposure, children showed no significant difference in AT association strength between the first half (trials 1–8) and the second half (trials 9–16) of AT test trials (see Supplemental Figure S4 from [14] Experiment 2).

### 1.3. Current Study

In the current study, we examined how the amount of prior experience influenced the strength of AT associations in typically sighted 6- to 8-year-olds. Children were provided with either four or eight trials of prior AV exposure. During exposure, they judged which of two abstract visual shapes matched a given nonsense sound, with no feedback provided. Following AV exposure, we assessed the strength of AT associations. Then, we considered how the strength of one type of crossmodal correspondence, AV, related to the strength of another type of crossmodal correspondence, AT, in a within-subject analysis that allowed us to account for individual differences. We expected that (1) AT association strength would increase with more versus less prior AV exposure, and (2) individuals performing better on AV trials would also perform better on AT trials.

## 2. Experiment 1: Four vs. Eight Trials of Prior AV Exposure

### 2.1. Methods

#### 2.1.1. Participants

A total of 61 children participated in the current study, providing usable data. The mean age of children was 7.50 years (range = 6.13–8.77) in the AV prior exposure condition with four trials and 7.57 years (range = 6.38–8.81) in the condition with eight trials (see Table 1 for demographics). Participants were recruited to come to our lab on the UMass Boston campus or through various museum venues in New England, including the Children’s Museum of New Hampshire (Dover, NH, USA), the Acton Discovery Museum (Acton, MA, USA), and the Children’s Museum in Easton (North Easton, MA, USA). All children gave verbal assent and written consent when possible, and parents or legal guardians gave written consent for their children to participate. The study, including all procedures and protocols, was approved by the Institutional Review Board of the University of Massachusetts Boston and complied with the Code of Ethics of the World Medical Association (Declaration of Helsinki).

#### 2.1.2. Stimuli and Apparatus

The auditory stimuli included four nonsense words—“baba”, “gaga”, “kiki”, and “titi”—which were recorded to be of uniform duration and amplitude. The visual stimuli included two pairs of abstract visual shapes, one pair with thick protrusions and one pair with thin protrusions. Within a given pair, shapes differed in whether the tips of the protrusions were round or spiky (see Figure 1A). These auditory and visual stimuli have been used in previous studies ([13]; [14]). Stimuli to be explored haptically consisted of 3-D shapes made using Tinkercad (https://www.tinkercad.com). A 3-D printer was used to create physical replicas of the visual stimuli, matching their two-dimensional outlines but rendered as flat shapes with a thickness of 1 cm (see Figure 1B). The maximum length across any pair of opposing protrusions was 10 cm. The surface of the shapes was not textured.

Auditory and visual stimulus presentation was controlled using MATLAB (R2014b or 2009b) with Psychtoolbox Version 3.0.12 ([7]; [40]) on a 15-inch MacBook Pro. Responses were collected via a Cedrus response pad (RB-844, Cedrus, San Pedro, CA, USA). Auditory stimuli were delivered through noise-canceling headphones (3M Peltor, HTB79A, Maplewood, MN, USA), and visual stimuli were displayed on the laptop screen. Tactile stimuli were presented inside a cardboard box. The front side, facing the participant, had two cloth-covered openings to block visual access. The back, open to the experimenter, enabled the placement and replacement of shapes between trials (see Figure 2 from [14]). A Logitech C920× HD Pro camera (frame rate = 30 frames per second, Logitech, Suzhou, China) recorded participants’ hand movements during haptic exploration.

#### 2.1.3. Procedures

Participants were randomly assigned to the 4- or the 8-trial condition. In the 4-trial condition, participants completed 4 AV trials (Exposure Phase) followed by 16 AT trials (Test Phase). In the 8-trial condition, participants completed 8 AV trials (Exposure Phase) followed by 16 AT trials (Test Phase). Practice trials were conducted prior to the exposure phase to verify children’s understanding of the task and to adjust sound levels as needed. An instructional video introduced contour following, an effective haptic strategy for exploring the shape contour, which involved an experimenter tracing the outline of familiar objects, such as a car or elephant (see Figure 2 from [10]). Children then practiced by haptically exploring the car and elephant shapes placed inside the cardboard box. Then, either a car sound or an elephant sound was played, and children had to judge which shape best matched the sound. Children had to complete 3 consecutive practice trials correctly. The data from children who failed to reach this criterion over 16 practice trials was excluded from further analysis.

During the Exposure Phase, participants sat in front of a laptop screen displaying a pair of shapes—one round and one spiky—while listening to a single auditory stimulus (“baba”, “gaga”, “kiki”, or “titi”; 700 ms each) through headphones. They were instructed to select the shape that best matched the sound by pressing a button on the corresponding side, either left or right. Trials with no response within 30 s were terminated and repeated. *Of note, the exposure phase provided no training since no feedback was given as to the response considered correct—feedback involved spinning around the shape the child selected to indicate the shape they had picked.*

During the Test Phase, participants sat in front of a cardboard box with two curtain-covered cutouts. An initial auditory signal prompted them to insert both hands into the left and right openings to explore the shapes via touch. While participants were exploring the shapes haptically, one of the four nonsense sound (“baba”, “gaga”, “kiki”, or “titi”) was presented via headphones (duration: 700 ms). Participants responded by indicating the side corresponding to the matching shape—either verbally, by tapping the chosen shape, or by raising the corresponding hand. The experimenter recorded the choice using a response pad. Afterward, a second tone signaled the end of the trial and cued participants to remove their hands from the box If no response was made within 30 s, the trial was aborted and repeated. Participants completed 16 trials in total (4 nonsense words × 2 pairs of shapes × 2 repeats). Hand movements were recorded throughout using a video camera. *Of note, the test phase provided no training, since no feedback was given as to the response considered correct—feedback involved spinning around the shape the child selected to indicate the shape they had picked.* See Figure 2 for experimental set-up for exposure and test phase.

#### 2.1.4. Measures

*Binary choice response (0 or 1).* In each trial, participants were given a pair of abstract shapes, either round or spiky, and asked to judge which shape best matched the nonsense sound they heard. A participant’s response for each trial was coded as a binary choice: 1 if they chose the expected, congruent, shape for a given sound, and 0 if they chose the opposite, incongruent, shape.

*Overall association strength (range = 0 to 1).* To measure the strength of the BK effect across trials, we calculated the overall association strength (Equation (1)) by dividing the total number of trials where participants chose the expected shape for a given sound (round for/a/vowel sounds and spiky for/i/vowel sounds) by the total number of trials completed. Higher association strength indicates more correct trials, with participants choosing congruent shape–sound pairs. If participants made their choice randomly, this proportion was 0.5.(1)$$Overall\;AT/AV\;association\;strength=\frac{number\;of\;trials\;with\;expected\;shape\;preference}{total\;number\;of\;trials}$$

### 2.2. Statistical Analyses

All analyses described below were performed in R ([42]).

*Presence/Absence of association*. We compared AT association strength against 0.5 to establish the presence of an association in each experimental condition using the t.test and wilcox.test functions. The *p* values were adjusted for multiple comparisons using the p.adjust function to control for false discovery rate ([5]). The effect size was reported as Hedges’ *g* (Cohen’s d corrected for small sample size) and estimated using the R package (Version 4.4.2, https://www.r-project.org/) “effsize” ([54]). Results were similar across *t*-tests and Wilcox tests, so *t*-test results were reported for easier comparison with previous literature.

*Magnitude of association*. We also modeled changes in the strength of AT associations applying a mixed-effect logistic regression model to participants’ binary choice response using the R package “lme4” ([4]) glmer function.

To compare the strength of the BK effect across exposure conditions, we performed mixed-effect logistic regression modeling to predict whether participants chose the abstract shape congruent to the nonsense sound presented, with the number of prior exposure trials (4 or 8) as fixed-effect factors, participant as a random effect (intercept), and a binomial distribution (Laplace approximation). We expected to see a significant effect of exposure group on the magnitude of AT associations, with more AV exposure resulting in stronger AT associations. The final model included 976 raw choice responses collected from 61 participants.

*Correlation in the Strength of AV and AT Associations*. We also considered the role of individual differences in learning during AV exposure trials in accounting for variability in subsequent AT test trials. Although the AV prior exposure phase did not provide explicit training, i.e., no feedback was provided, children may have been learning what features of visual shapes and/or sounds to pay attention to and how to compare and contrast features, with more trials, i.e., exposure, resulting in more learning and stronger AT association strength during test trials. Individual differences in what may have been learned during the AV exposure phase, with better learning reflected in better AV performance and worse learning reflected in weaker AV performance, could yield subsequent better and worse AT performance during the test phase, respectively. To consider this possibility, we examined the correlation between AV association strength during the prior exposure phase and AT association strength during the subsequent test phase on a subject-by-subject basis. We expected participants showing stronger AV associations during the exposure phase to also show stronger AT associations during the test phase.

We analyzed data from Experiment 1 of the current study (4 and 8 trials of prior AV exposure) and data from Experiment 2 of [14] ([14]) (16 trials of prior AV exposure). We quantified the correlation between the magnitude of the overall AT association strength for test trials and the magnitude of the overall AV association strength for exposure trials, computing each measure for each participant. Because overall AT association strength was not normally distributed (*W*s ≤ 0.9749, *p*s < 0.011), we performed non-parametric correlational analyses and reported Kendall’s tau and Spearman’s rho, which are interpreted in a similar way as Pearson’s *r*.

### 2.3. Results

Our final sample of usable data consisted of 61 participants: 30 participants completed the 4-trial condition, with an additional 7 participants excluded because they did not reach criterion during practice trials (n = 4), did not complete all 16 trials of the test phase (n = 2), or did not understand the instructions (n = 1). A total of 31 participants completed the 8-trial condition, with an additional 10 participants excluded because they did not reach criterion during practice trials (n = 1), did not complete all 16 trials of the test phase (n = 6), did not understand the instructions (n = 1), saw the shape during the AT task (n = 1), or had developmental concerns (n = 1).

#### 2.3.1. Presence/Absence of AT Associations

Figure 3 plots the average AT association strength (±sem) for 6- to 8-year-olds completing 4 or 8 AV trials during the exposure phase of the current study. For context and comparison, we also include data from a related study in 6- to 8-year-olds using the same paradigm with children completing either 0 or 16 AV trials during the exposure phase ([14]: Experiment 2). AT association strength was not significantly different from chance, or 0.5, in participants first completing 0 trials (*t*(14) = −0.11, *p* = 0.91, *g* = 3.384) or 4 trials of AV exposure (*t*(29) = 1.78, *p* = 0.1, *g* = 2.765). Unlike the results with fewer trials, AT association strength was significantly different from chance in participants who first completed 8 trials (*t*(30) = 5.536, *p* < 0.001, *g* = 3.618) or 16 trials of AV exposure (*t*(16) = 3.454, *p* < 0.001, *g* = 2.618). Importantly, not only are *p* values significantly different from chance, but effect sizes are robust, above values considered large for Hedge’s g, where a large effect size would be 0.8 or above.

#### 2.3.2. AT Association Strength with Differing Amounts of Prior AV Exposure

To quantify changes in the magnitude of AT associations as a function of AV exposure, we performed a mixed-effect binary/logistic regression to predict participants’ odds of choosing the expected shape, making a congruent response, with a given amount of AV exposure (4 or 8 trials). The alternative model (Akaike information criterion [AIC] = 1230.0, Bayesian information criterion [BIC] = 1244.6, log likelihood = −611.99), including AV exposure and interaction terms with AV exposure, significantly improved model fit (*x*^2^(1) = 5.724, *p* = 0.017) compared with the null model (AIC = 1233.7, BIC = 1243.5, log likelihood = −614.86; see Appendix A for the null model and a comparison of model fit, respectively). Table 2 lists the odds ratio coefficient of each predictor in the alternative model. For AV exposure, the results showed that the odds of making a correct response increased by a factor of 85.4% when participants were given 8 trials versus 4 trials of AV exposure (*p* = 0.015). The odds ratio of 1.854 is considered a small to medium effect size.

#### 2.3.3. Correlation of AV and AT Association Strength

Figure 4 plots the strength of the overall association strength for each individual participant during the Exposure Phase (AV) and during the subsequent Test Phase (AT). The strength of AT associations during the test phase increased with increasing strength of AV associations during the exposure phase (*tau* = 0.412, *z* = 4.900, *p* < 0.001; *rho* = 0.529, *S* = 38733, *p* < 0.001). Importantly, *p* values are significantly different from chance, and effect sizes (Kendall’s tau and Spearman’s rho) are also robust, falling between small and medium values, where a medium effect size is 0.5. These findings suggest that individuals who are better on one type of BK correspondence—audio–visual—are also better on another type of BK correspondence—audio–tactile.

## 3. General Discussion

While many studies have considered the development of Bouba–Kiki (BK) associations between nonsense sounds and abstract visual shapes (AV: e.g., [35]; [38]; [22]; [13]; [46]), much less is understood about the complementary development of BK associations between nonsense sounds and abstract shapes only experienced via touch (AT). Furthermore, few studies have considered the relationship between these two distinct forms of sound–shape crossmodal correspondence: AV and AT. Previous studies have highlighted the importance of early visual experience in forming AT associations by considering evidence in blind adults (e.g., [23]; [26]; [47]) as well as typically sighted children ([14]), whose visual experience and capacity for visual imagery and visual representation are still naïve ([37]).

The goal of the current study was to examine how the *amount* of prior AV experience influenced subsequent AT associations early in development. To examine this, we extended previous work that showed 16 trials of prior AV exposure enhanced AT associations in children 6 to 8 years old. Children completing a simple AV matching task (indicating which visual 2-D shape displayed on a screen, spiky or round, best matched a nonsense word) were better at indicating which 3-D shape (spiky or round, manually explored without visual feedback) best matched a nonsense sound, a complementary AT matching task ([14]). Here, we tested whether four or eight trials of prior AV exposure was also sufficient to enhance AT associations. We expected that increased AV exposure would enhance AT association strength. In line with our prediction, our results suggest that while eight trials of prior AV exposure can enhance AT association strength, four trials is not sufficient. We also considered how individual differences influenced the strength of AT associations. We expected that individuals who performed better on AV trials would also perform better on AT trials. In line with our prediction, we found that children who performed better during the AV matching task also performed better on the subsequent AT matching task. Future work would need to determine if such effects reflect a mechanism that allows for gradual strengthening of associations across trials or involves a threshold or break point.

### 3.1. Is Enhanced AT Association Due to Repeated Testing?

What could account for the increases we observed in AT association strength? One may argue that the enhanced AT association effect we observed here is a result of practice or time-on-task for the AT task itself. To rule out this possibility, we compared the AT association strength with four versus eight trials of prior AT exposure, as AT prior exposure does not include visual images of shapes, as found with AV prior exposure. We re-analyzed data from [14]’s ([14]) Experiment 1, in which a different cohort of 6- to 8-year-olds completed 32 AT trials. We did not find evidence of strengthened AT associations with practice: neither 4, 8, nor 16 trials of prior AT exposure strengthened subsequent AT associations in children (see Appendix A). Additionally, using the same trial-by-trial analysis conducted by [26] ([26]), we found that children showed no significant AT association in trials 1, 4, 5, or 8 (see Appendix A). Thus, our findings in 6- to 8-year-olds differ from findings in blind adults, who show enhanced AT performance on later trials (4, 5, and 8) and from blindfolded sighted-adults who show a deterioration in performance in later trials (4, 5, and 8; [26]). Overall, the amount of exposure, i.e., the number of trials, is a critical factor to consider in studying the BK effect. Ultimately, the mechanisms preventing young children from showing stronger associations with repeated AT exposure differ from those that allow blind adults to exhibit stronger associations with repeated AT exposure or sighted adults, when blindfolded, to exhibit weaker associations with repeated AT exposure.

### 3.2. What Is Being Learned During AV Exposure?

The observation that stronger AV performance correlates with stronger AT performance suggests that children are learning during AV exposure trials. Such learning is not due to explicit feedback, i.e., children are not told which responses are correct, and learning is specific to AV exposure, as additional AT exposure has no benefit. Yet, what is being learned during AV exposure? While a review of unisensory and multisensory learning is beyond the scope of this paper (e.g., [44]; [36]), we suggest two main possibilities.

One possibility is that during exposure to AV stimuli, children focus on visual information. Seeing a pair of visual 2-D shapes, which differ only in protrusion tip, i.e., round vs. spiky, could aid in the subsequent processing of complementary tactile 3-D objects with identical shape features. The visual display highlights the contrast in shape contours, round vs. spiky, and could enhance the child’s ability to form a visual mental image when they touch similar tactile shapes and/or could enhance the child’s ability to explore tactile shapes more effectively when shapes are only touched but not seen. *With repeated AV exposure, the reliability and/or vividness of the visual mental image could be strengthened, or the understanding of what features of the shape are important could be enhanced*. More reliable visual imagery or a clearer understanding of relevant shape features during AV exposure could help children call to mind the visual mental image or image features (round or spiky protrusions) when they only touch similar shapes during the AT test, allowing for stronger performance on the AT task when performance is stronger on the AV task.

Another possibility is that AV exposure allows children to better understand how the nonsense sounds that they hear relate to the shapes that they see. *With repeated AV exposure, the correspondence between sounds and shapes could become clearer.* Hearing the same sounds during subsequent AT trials might allow children to call to mind the sound–shape correspondence they selected during AV exposure, allowing for stronger performance on the AT task when performance is stronger on the AV task.

The current experimental design and results cannot tease apart the possibilities highlighted here, nor can they address individual differences arising from other cognitive factors, such as differences in visual or auditory short-term or working memory. Future work would need to directly test between possible explanations and underlying mechanisms. For example, to consider the first possibility outlined above, visual imagery could be quantified explicitly by having children draw what they imagine. To consider the second possibility, only visual images could be presented during prior exposure to determine if sounds play an important role.

### 3.3. The Development of Haptic Processing

In our study, haptic exploration required participants to determine the contour of the shapes they were touching. The shapes we used were abstract, 3-D, flattened objects of uniform thickness, rigidity, and texture, only differing in shape contour, with round or spiky protrusions. Overall, we found that AT associations for these shapes were weak but modifiable by select experience, benefiting from prior AV exposure. A benefit from prior AV exposure would be expected if haptic exploration of our abstract shapes is challenging and might require the ability to create a visual mental image of the shape being explored via touch. Direct visual input, seeing a visual depiction of the object, could help a child better imagine what they are feeling, create a visual mental image, and optimize the strategies used to explore objects manually. In fact, the *image mediation model* of haptic processing, first proposed by [31] ([31]), suggests that the ability to transfer haptic inputs into a visual mental image might be important for recognizing objects explored manually. The image mediation model might be preferentially invoked in processing the flattened 3-D abstract shapes used in our study, and the select prior AV exposure may facilitate translating what is explored haptically into a visual mental image.

Translating information obtained via touch to a visual depiction of the object, *haptic-to-visual transfer,* is an ability that emerges slowly over the course of development. Infants as young as 4 months of age can translate what they learn via touch by manually manipulating two rings hidden from sight, and then prefer to look at the corresponding visual depiction of the two rings connected rigidly or flexibly ([51]). However, for finer, detailed representations, 5-month-old infants were unable to transfer shape and texture information acquired via touch onto a visual depiction of the object, i.e., habituation did not transfer from touch to vision, although infants could transfer information from vision to touch ([50]). Even 5-year-olds have difficulty recognizing an unfamiliar object explored only via touch from an array of visual prototypes ([9]; [43]; [29]). Interestingly, priming 4-year-olds with 8 trials of a visual image of an unfamiliar object enabled them to better identify a visual prototype after manually exploring an array of unfamiliar objects ([28]). Thus, prior visual experience can facilitate haptic-to-visual transfer. A similar mechanism may be engaged in our task when prior AV experience enhances subsequent AT associations. Overall, optimal integration of visual and haptic cues requires extensive calibration between sensory systems (e.g., judging object size: [24]), not reaching adult-like optimal integration until 8–10 years of age and often depending on visual experience (e.g., limited object orientation discrimination by touch in congenitally blind children: [25]).

The slow development of haptic exploration itself might also contribute to the slow development of haptic-to-visual transfer. According to [33] ([33]), haptic object exploration is an iterative process involving selection and extraction. In each cycle, a specific exploratory procedure is applied to gather information about a particular dimension of the object. The choice of strategy during shape exploration highlights how specific haptic features are actively identified and interpreted. Haptic exploration is complex, relying on various skills, such as visuo-spatial skills (e.g., [34]), which emerge with experience. Children’s use of strategies is sub-optimal. For example, given the same instructions in a match-to-sample task focusing only on haptic information (shape, texture, weight, and volume), although 7- to 12-year-olds used the same haptic exploration strategies as adults, they also used additional suboptimal strategies. This suggests children are less proficient in their use of haptic exploration strategies; however, in this haptic-only task, the use of haptic strategies improved with practice ([56]). In related studies where AV prior exposure enhanced AT associations, haptic exploration strategies became more optimal, with increased use of more effective strategies (i.e., poking the tip of the shape) and decreased use of ineffective strategies (i.e., sweeping the surface of the shape) ([10]).

## 4. Conclusions

Previous research suggests that cross-modal associations between how an abstract shape feels and the sounds corresponding to the shape (AT sound–shape correspondence) are weak or absent in individuals with limited or naive visual experience, including in congenitally blind adults and typically sighted children, respectively. AT associations can be enhanced in children who first see abstract visual shapes resembling the touched shapes ([14]). Here, we examined how much prior experience is sufficient to strengthen AT association in early development. We found that eight but not four trials of prior AV experience can enhance AT associations in 6- to 8-year-olds. Furthermore, stronger AV associations correlate with stronger AT associations, suggesting that individual differences in what children may be learning during exposure are important to consider.

## Figures and Tables

**Figure 1 behavsci-15-01184-f001:**
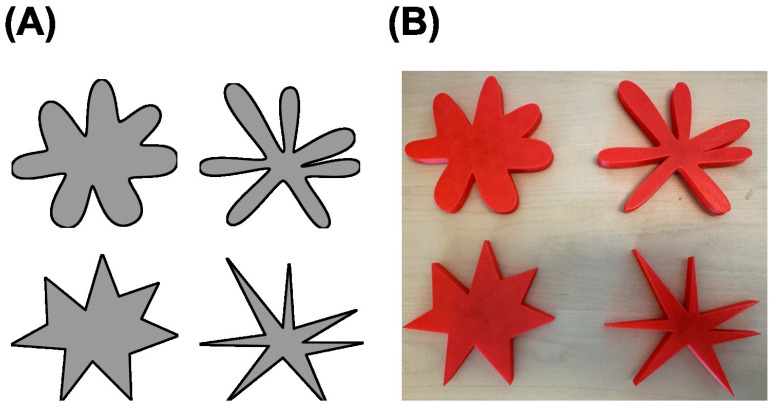
Two-dimensional (**A**) and three-dimensional (**B**) abstract shape pairs used for visual and haptic exploration. Each pair included one round and one spiky shape, with the top pair having thicker protrusions and the bottom pair having thinner protrusions. Protrusion size was held constant across all conditions; only the contour spikiness, round versus spiky, of the shape varied between the paired stimuli (adapted from [10]).

**Figure 2 behavsci-15-01184-f002:**
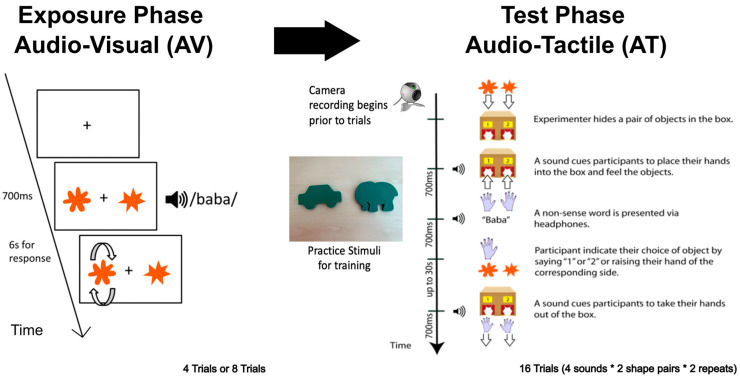
Experimental procedures. Participants were randomly assigned to either 4 trials or 8 trials of AV prior exposure. During the AV exposure phase, participants were asked to pick the visual shape shown on a computer screen that best matched the nonsense sound they heard. After the exposure phase, participants completed 16 trials of AT sound–shape correspondence. For each AT trial, a sound cued participants to place their hands in the box, and then participants heard a nonsense sound (700 ms)—“baba”, “gaga”, “titi”, or “kiki”—while feeling two opposing tactile shape pairs. Participants had up to 30 s to select which shape best matched the sound. After participants indicated their choice verbally or by raising their hand on the side corresponding to where they felt the matching shape, a sound cued them to take their hands out of the box. Each participant completed a total of 16 trials (4 nonsense words × 2 pairs of shapes × 2 repeats).

**Figure 3 behavsci-15-01184-f003:**
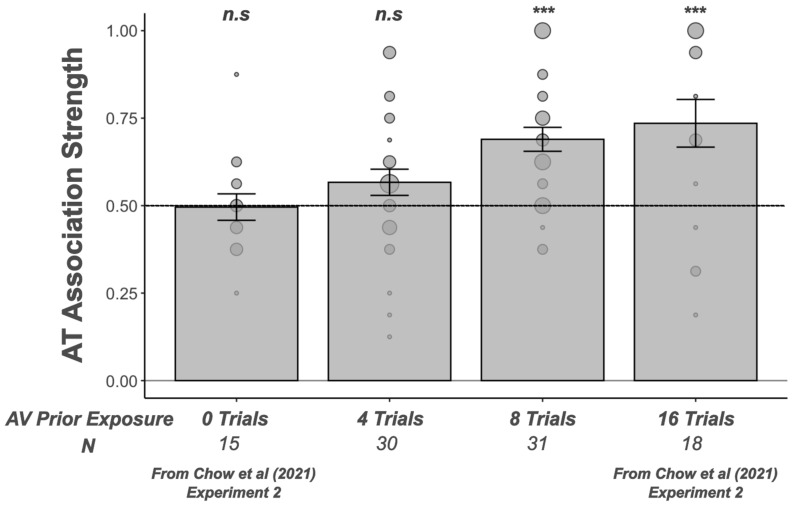
Audio–tactile (AT) association strength for 0, 4, 8, and 16 trials of AV exposure. Individual AT association strength (dots with size scaled to account for number of participants at the same value) and mean AT association strength (bar plot ± standard errors across participants) results show that children with 4 trials of AV exposure did not associate nonsense words and abstract tactile shapes, whereas children with 8 trials of AV exposure showed a significant association between nonsense words and abstract tactile shapes. A subset of data from a different group of participants tested in Experiment 2 of [14] ([14]) also showed significant AT association following 16 trials of prior AV exposure, but not 0 trials of prior AV exposure. All participants completed 16 AT trials. N: number of participants; *n.s.*: nonsignificant; *** *p* < 0.001.

**Figure 4 behavsci-15-01184-f004:**
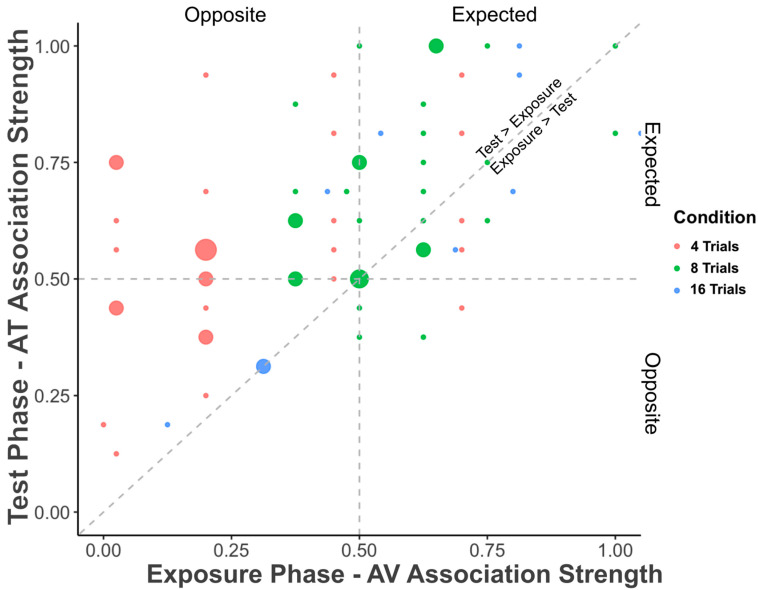
Relationship between the overall AV association strength during the exposure phase and AT association strength during the test phase as a function of number of trials of AV exposure. Individual association strengths are depicted for each participant by dots, which are scaled in size to account for the number of participants at the same value, with color indicating the number of exposure trials (red: 4 trials; green: 8 trials; blue: 16 trials). Participants showing stronger AV associations during the exposure phase were also more likely to show stronger AT association during the test phase.

**Table 1 behavsci-15-01184-t001:** Participant demographics.

Condition	n	Mean Age	% Female	% Right-Handed	% Identifying as White	% English as (One of) First Language(s)	% Bilingual
Four Trials	30	7.50	63.33	86.67	73.33	86.67	23.33
Eight Trials	31	7.57	41.94	83.87	77.42	90.32	35.48

**Table 2 behavsci-15-01184-t002:** Fixed effects of the final mixed-effect logistic regression model predicting binary response using number of AV exposure trials as predictors.

Fixed Effects	Coefficient	Odd Ratio	SE	*z* Value	*p* Value	95% Confidence Interval	
						Lower	Upper
(Intercept)	0.310	1.363	0.178	1.737	0.082	−0.045	0.672
AV exposure							
1: 8 Trials	0.617	1.854	0.254	2.431	0.015	0.116	1.138
0: 4 Trials							

## Data Availability

The dataset generated for this study has been made available on the Open Science Framework: https://osf.io/kh8d7/?view_only=1c718ce1bf1544e2a7e6b0149f511ad6 (accessed on 1 March 2025).

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
