# Peer review of "Four Trials Is Not Enough: The Amount of Prior Audio–Visual Exposure Determines the Strength of Audio–Tactile Crossmodal Correspondence Early in Development"

_behavsci, 2025, doi:10.3390/bs15091184_

Round 1

Reviewer 1 Report

Comments and Suggestions for Authors

This article investigates whether exposure to audio-visual stimulus matching tasks enhances the ability to associate auditory and haptic stimuli in children ages 6-8. Previous research (Chow 2021 et al., referenced in this study) has provided evidence that this is indeed the case; the present article seeks to establish how much exposure is needed for this effect to obtain.

The relevance of the article is made very clear by its embedding into broader research concerning multisensory processing and the so-called 'bouba/kiki' effect. The main experiment is well designed and the results are instructive. 

However, I have two main reservations with regard to this submission.

First, the article seems to cover much of the same ground as the study by Chow et al. (2021); one of the authors of the present submission was also a co-author of that paper. At 349, the authors refer to 'our previous results', citing Chow et al. (2021). The results of that study are incorporated into the present submission at several points. Moreover, there is also a very recent study by Cao et al. (2024), including authors of the present article, that again makes a very similar contribution (the first sentences of that study very closely resemble the first sentences of the abstract here). All in all, the present study seems to be an expansion of this earlier research, which raises the question whether it justifies an article of this length.  

Second, the authors' approach to statistical significance is black-and-white, and this impairs the conclusions they draw from their data. Consider figure 3, representing the results from the main experiment. The figure suggests that, as the number of trials increases, the association strength also increases. However, because for 4 trials, the associated p-values stay below the arbitrary threshold of 0.05, the authors conclude that 4 trials are 'not enough' to have an effect. This suggests that somewhere between 4 and 8 trials, there is some magical boundary after which the trials suddenly have an effect. In reality, the evidence suggests a more continuous correlation. Note also that the odds ratio between the effects for 4 and 8 trials is reported as being 1.180 (342), which points to only a slight effect. Given that previous research has already established an effect after 16 trials, the question is what the present research really adds; in my view, it merely shows that, the more trials, the stronger the effect, which is worth noting but not very surprising.

The same point applies to figure 4, although to a lesser degree. Here too, there seems to be a consistent tendency for the association strength to increase. It is true that the effect is quite subtle, but at the same time, there were only 20 participants here, so it is much harder to obtain a 'significant' p-value. Also, as a 'control' study, this secondary analysis, in my view, fails to have its intended effect. For even if there were a clear correlation between the number of AT trials and the subsequent association strength (also during AT trials), that would not necessarily invalidate the hypothesis that AV trials also increase the subsequent association strength. It is possible that they both have a similar effect, but for different reasons. Instead, a more interesting control study, I believe, would be the one suggested in 664-666: to contrast the condition in which AV trials are contrasted with mere exposure to visual stimuli. 

Detailed comments:

Title: ‘the strength of audio-tacticle crossmodal correspondences’ – this suggests that it is the correspondences themselves that are strenghtened, but what is strenghened is rather the ability to integrate multisensory signals. 

13 ‘objects only explored via touch’ – presumably, 'where the objects are only explored via touch'. 

14-14 ‘with more naïve visual experience’ – I only understood that the relative clause was non-restrictive after reading lines 113-114. That is, the authors do not intend to designate a subset of these children, but rather signal that these children all have more naïve visual experience. It would be clearer to write ‘who have’ following a comma.

15 ‘Interestingly’ – is this remarkable, given the previous statement that visual experience influences AT associations? Perhaps simply 'It has been found ...'?

34-36 ‘spatially or temporally congruent … appear from the same spatial location … spatially co-localized’ – this seems redundant.

48 ‘baluma’ – I believe the word is maluma (the b may be due to conflation with bouba?).

119 ‘only 16 trials of an AV BK task’ – I find the apposition hard to integrate into the larger sentence. Also, it is not clear why 16 would necessarily be a low number.

132 ‘when a new shape pair was introduced’ – the experiment is said to have featured raised outlines as well as raised filled-in forms. This raises the question what the effect of the variable outline/filled-in was. As it stands now, it can be a bit confusing to understand what exactly was at stake in the experiment.

133-134 ‘association strength diminished’ – if the blindfolded participants actually got worse at the task, then how does this relate to the following claim that ‘AT associations were strengthened when additional AT exposure provided extra information’? Or does ‘extra information’ imply that this applied only to the blind participants (because for those with sight, the information was not extra)?

139 ‘Of note …’ – has this not already been said in 107 and 124-125? 

151 ‘the amount of prior experience’ – here the reference seems to be specifically to visual experience (AV exposure), given the initial description of the task. However, later a control analysis of AT exposure is announced. I think it should be made much more clear from the outset that the results of different studies will be integrated. 

170 ‘61 children’ – it may be useful here to note the total number of children who tried to do the experiment.

272 ‘a given vowel’ – I would highlight the fact that the two conditions also differ in the voicing of the semivowels. 

342 ‘1.180' - in the table, the odds ratio is reported as 1.117. It seems that the table does not report the odds ratio but the regression coefficient; see 418 ('odds ratio coefficient). 

Author Response

We thank Reviewer 1 for their careful review of our work and for the detailed and constructive comments. Below please find a point-by-point reply showing how we addressed Reviewer 1's concerns.

DETAILED COMMENTS:

Title: ‘the strength of audio-tacticle crossmodal correspondences’ – this suggests that it is the correspondences themselves that are strenghtened, but what is strenghened is rather the ability to integrate multisensory signals. 

We thank Reviewer 1 for making this point, which is important. Throughout our manuscript, we have remained agnostic as to the mechanisms involved in crossmodal correspondences and have avoided using the word “integration”. It is not clear to us what mechanism may be involved both because of differences presented in the literature and from feedback we’ve received when presenting our data at conferences.

13 ‘objects only explored via touch’ – presumably, 'where the objects are only explored via touch'. 

We thank Reviewer 1 for this suggestion. We have revised the sentence, as suggested, to say: “where the objects are only explored via touch”

14-14 ‘with more naïve visual experience’ – I only understood that the relative clause was non-restrictive after reading lines 113-114. That is, the authors do not intend to designate a subset of these children, but rather signal that these children all have more naïve visual experience. It would be clearer to write ‘who have’ following a comma.

We thank Reviewer 1 for this helpful suggestion to make this sentence clearer. We have revised this sentence to read: “Visual experience influences AT associations, which are weak in early-blind adults and in fully-sighted 6- to 8-year-olds, who have more naïve visual experience”

15 ‘Interestingly’ – is this remarkable, given the previous statement that visual experience influences AT associations? Perhaps simply 'It has been found ...'?

We thank Reviewer 1 for this comment. We have revised the sentence to “It has been found that prior AV exposure in children enhances AT associations.”

34-36 ‘spatially or temporally congruent … appear from the same spatial location … spatially co-localized’ – this seems redundant.

We thank Reviewer 1 for this comment. We have revised the sentence to “Multisensory signals are more likely to arise from a common source and be integrated if they are either spatially or temporally congruent, meaning they appear from the same spatial location, or change in the same way over time (i.e. Talsma & Woldorff, 2005; Stein & Stanford, 2008; Talsma et al., 2010; Chen & Spence, 2017).”

48 ‘baluma’ – I believe the word is maluma (the may be due to conflation with bouba?).

We thank Reviewer 1 for highlighting this error. We have revised the word to “maluma”.

119 ‘only 16 trials of an AV BK task’ – I find the apposition hard to integrate into the larger sentence. Also, it is not clear why 16 would necessarily be a low number.

We have removed the word “only” and the word BK, so that this sentence is hopefully clearer, that AV prior exposure influences subsequent AT associations.

132 ‘when a new shape pair was introduced’ – the experiment is said to have featured raised outlines as well as raised filled-in forms. This raises the question what the effect of the variable outline/filled-in was. As it stands now, it can be a bit confusing to understand what exactly was at stake in the experiment.

We thank Reviewer 1 for this comment. In Graven and Desebrock (2018), the experimental design used identical pairs of 2-D raised-line drawing of abstract shapes which were outlined or filled in. Participants completed 4 trials of one type of shape pair, i.e., outlined, followed by 4 trials of the other type of shape pair., i.e., filled They compared 3 group of participants: blindfolded sighted, sighted, and early-blind adults. They argue that if a certain group of participants showed an immediate AT association, more than 50% of the participants in a particular group would correctly match the nonsense sound with one of the two abstract shapes felt by hand in Trial 1. If AT association emerged with repeated testing of the SAME pair of abstract shapes, they expected more than 50% of the participants in a particular group would correctly match the nonsense sound with one of the two abstract shapes felt by hand in Trial 4. Similarly, the effect of repeated testing of a DIFFERENT pair of abstract shapes were expected to emerge in Trial 5 and/or Trial 8. Although our study did not use raised line drawings and did not vary shape pairs based on filled or outlined, we varied protrusion thickness randomly, the main point is changes in the association strength with practice.

133-134 ‘association strength diminished’ – if the blindfolded participants actually got worse at the task, then how does this relate to the following claim that ‘AT associations were strengthened when additional AT exposure provided extra information’? Or does ‘extra information’ imply that this applied only to the blind participants (because for those with sight, the information was not extra)?

We thank Reviewer 1 for highlighting this confusion. Graven and Desebrock’s work (2018) argues that prior exposure helps AT associations only when it comes from the dominant intact sense. Thus, early blind participants, with deprived vision, rely heavily on the haptic sense to process sensory signals, meaning prior exposure of abstract shapes through touch helps them make better AT association. On the other hand, blindfolded sighted participants, rely heavily on the visual sense to process sensory signals, suggesting prior exposure of abstract shapes through touch does NOT help them make better AT association. We have now added wording for clarity: “Overall, these effects of time-on-task suggest that AT associations were strengthened when additional AT exposure, 4 trials, provided extra information in the intact sensory system that participants tend to rely on.”

139 ‘Of note …’ – has this not already been said in 107 and 124-125? 

We thank Reviewer 1 for highlighting this redundancy - we have now deleted this sentence to reduce redundancy.

151 ‘the amount of prior experience’ – here the reference seems to be specifically to visual experience (AV exposure), given the initial description of the task. However, later a control analysis of AT exposure is announced. I think it should be made much more clear from the outset that the results of different studies will be integrated. 

We thank Reviewer 1 for this comment. We have now altered the manuscript and included the control analysis of AT exposure in supplemental material, as suggested by other reviewers. Hopefully this helps better focus our main results.

170 ‘61 children’ – it may be useful here to note the total number of children who tried to do the experiment.

We thank Reviewer 1 for this comment. The information about additional excluded participants can be found in the beginning of the Results section (see section 2.2): “Our final sample of usable data consisted of 61 participants: 30 participants completed the 4-trial condition, with an additional 7 participants excluded because they did not reach criterion during practice trials (n=4), did not complete all 16 trials of the test phase (n=2), or did not understand the instructions (n=1). 31 participants completed the 8-trial condition, with an additional 10 participants excluded because they did not reach criterion during practice trials (n=1), did not complete all 16 trials of the test phase (n=6), did not understand the instructions (n=1), saw the shape during the AT task (n=1), or had developmental concerns (n=1).”  

272 ‘a given vowel’ – I would highlight the fact that the two conditions also differ in the voicing of the semivowels. 

We thank Reviewer 1 for this comment - Per Reviewer 2’s request, we have redefined our measure in the section “choice bias separated by sound type” to present overall AT association strength in the congruent direction, defined by the percentage of trials participants made a correct sound-shape matching (i.e. choosing the abstract shape with round contour when an /a/-related vowel sound is played). Thus, we are not including mention of vowel voicing differences between conditions.

342 ‘1.180' - in the table, the odds ratio is reported as 1.117. It seems that the table does not report the odds ratio but the regression coefficient; see 418 ('odds ratio coefficient). 

We thank Reviewer 1 for this comment. We have fixed this typo in the manuscript: “The model reports a significant interaction between sound vowel category and number of AV exposure trials; the odds of choosing a round shape when an /a/ sound was presented increased by a factor of 1.117 (p < .001) when participants were given 8 trials of AV exposure versus 4 trials of AV exposure, holding other factors constant.”

Reviewer 2 Report

Comments and Suggestions for Authors

SUMMARY OF MANUSCRIPT
There is some question of the age at which auditory (A) sounds are associated with visual (V, 2D) and tactile (T, 3D) objects. Of interest to the current study is the age at which softer sounds such as baba or gaga are associated with rounded objects and percussive sounds such as kiki and titi with angular or spiky objects, and whether the amount of exposure to AV stimuli (4 vs. 8 trials) would (a) transfer to AT stimuli and (b) impact the strength of AT associations in 6-8 year children. Four trials were not enough to see the associations above chance level. However, with 8 trials, there was transfer from AV to AT, with effect sizes of g = 1.039 for soft/rounded stimuli and g = 1.54 for percussive/spiky stimuli [converted from effect size of -.614 in text]. The main conclusion is that 6–8-year-olds can exhibit this association if given enough trials.

EVALUATION OF MANUSCRIPT
The current manuscript has its strengths and weaknesses. The strengths are that the introduction was well structured and led clearly to the need for and design of the current study. The design is rigorous and affords testing of the hypotheses.

The weaknesses are that the analyses are more complicated than they need to be, and the secondary analyses do not add to the value of the current study; that is, they focus attention away from the current study.

My overall recommendation is to present only the current study and put the secondary analyses in a separate paper. However, the results for the current study need clarification and simplification. My specific comments are below.

Any questions I raise below are questions that readers will have and therefore the answers should be in the text and not just in the response to my review.

1. In the first analysis (presence/absence of AT association), the vowels were analyzed separately and yet the strength of the associations were calculated as if they were on the same scale (-.5 to .5, with 0 = chance).

AT effect for soft/round stimuli: p(round) = #round responses/#round trials - 0.5
Example: perfect AT score for round stimuli = 8/8 – 0.5 = +0.5

AT effect for percussive/spiky stimuli: p(round) = #round responses/#spiky trials – 0.5
Example: perfect AT score for spiky stimuli = 0/8 – 0.5 = -0.5

No AT effect = 4/8 – 0.5 = 0

Though these calculations present beautifully in Figure 3, they are misleading because they imply that the vowel categories were part of a single analysis when in fact each vowel category was tested separately (i.e., two t-tests).

It would be just as easy to present the data as the extent to which congruity was present. Congruity for soft sounds would be the proportion of trials on which responses were congruent: rounded objects were chosen for soft sounds and spiky objects were chosen for percussive sounds. This way, it would be clear what “strength of association” meant – higher numbers mean stronger associations. This is in contrast with the current calculations—values close to +0.5 for round objects and -0.5 for spiky objects are strong associations.

Note that a response of “that is the way the data have been analyzed and published before” is not a good reason for continuing to present results that are difficult to comprehend.

Please add a new heading in the results section, 2.1 Presence/Absence of AT Associations.

2. The logistic regression was intended to analyze the strength of the AT associations by AV exposure (4 vs. 8 trials), vowel category (soft vs. percussive) and their interaction. The DV was whether participants responded “round” (=1) on each AT trial. Although the equations discussed above were not used in this analysis, the logic is similar.

If I am understanding correctly, for the soft vowel category, values near 1 represent stronger associations (i.e., round objects were always chosen for soft sounds). However, for the percussive category, values near 0 represent stronger associations (i.e., round objects were never chosen for percussive sounds). Or is the focus of the analysis only on soft category sounds? What does “strength of association” refer to? Be specific.

Also, I am assuming that the odds ratios in Table 2 are log odds coefficients because odds ratios cannot be negative. According to my calculations, the odds ratios are:
Intercept = .79
AV exposure = .64
Sound category = 1.77
Interaction = 3.06

Nevertheless, multilevel logistic regression is still new to many people, so it is important to be clear about what the results mean. Unfortunately, the authors spend little time on the regression results. Here is one way to describe the results clearly yet succinctly.

For the AV exposure main effect (p = .049): The results show that responding to the sound with a round object is 36% less likely to occur after 8 trials (=1) than after 4 trials (=0).
Note: I calculated 36% using the formula % = (1-.64)*100
Note: this does not make any sense – what does it say about strength of association?

For the Sound category main effect (p = .003): Responding round is 77% (1.77) more likely for soft sounds than for percussive sounds.
Note: What does this mean in terms of strength of association between soft/round and percussive/spiky?

For the interaction (p < .001): The difference in responding round between soft and percussive sounds is 3.06 more likely after 8 trials than after 4 trials.
Note: The authors state that the strength of association is greater at 8 trials than at 4 trials. Is this for both soft/round and percussive/spiky stimuli?

3. I return to my comment in #1 about recoding the binary data. The descriptions would be much simpler. These following statements are not based on actual findings, but you can see the difference:

AT responses are more congruent after 8 trials than after 4 trials. (associations are stronger after 8 trials compared to 4 trials)

AT responses are more congruent for soft sounds than for percussive sounds (associations are stronger for soft sounds than for percussive sounds)

The difference in AT congruence (strength of association) between soft and percussive sounds is greater after 8 trials compared to 4 trials (or something like that).

4. I would like to see the results for all levels of the models: intercept only, CIM and/or AIM (intercept, main effects) and Interaction (intercept, main effects, and interaction), including a comparison of model fits. The model fits for the intercept only and final models are in the text, but I would like to see the model with main effects separately from the final model.

5. I recognize the value of the secondary analyses, but they complicate the straight-forward finds of the current study. They deserve to stand on their own. Yes, removing them would make the current study a short report, but that is ok.

6. I have not commented on the discussion because it may change extensively.

Author Response

We thank Reviewer 2 for their careful review of our work and for the detailed and constructive comments. Below please find a point-by-point reply showing how we addressed Reviewer 1's concerns.

DETAILED COMMENTS:

Please add a new heading in the results section, 2.1 Presence/Absence of AT Associations.

We thank Reviewer 2 for this insightful comment, which we agree would help us present our data more clearly and in a more focused manner. We have redefined the measure for “Overall AT association strength” to calculate the effect of congruency and have revised the data visualization and statistics reporting accordingly. In addition, we also added a new heading in the result section “2.2.1 Presence/Absence of AT association”

  1. The logistic regression was intended to analyze the strength of the AT associations by AV exposure (4 vs. 8 trials), vowel category (soft vs. percussive) and their interaction. The DV was whether participants responded “round” (=1) on each AT trial. Although the equations discussed above were not used in this analysis, the logic is similar.

If I am understanding correctly, for the soft vowel category, values near 1 represent stronger associations (i.e., round objects were always chosen for soft sounds). However, for the percussive category, values near 0 represent stronger associations (i.e., round objects were never chosen for percussive sounds). Or is the focus of the analysis only on soft category sounds? What does “strength of association” refer to? Be specific.

Also, I am assuming that the odds ratios in Table 2 are log odds coefficients because odds ratios cannot be negative. According to my calculations, the odds ratios are:
Intercept = .79
AV exposure = .64
Sound category = 1.77
Interaction = 3.06

Nevertheless, multilevel logistic regression is still new to many people, so it is important to be clear about what the results mean. Unfortunately, the authors spend little time on the regression results. Here is one way to describe the results clearly yet succinctly.

For the AV exposure main effect (p = .049): The results show that responding to the sound with a round object is 36% less likely to occur after 8 trials (=1) than after 4 trials (=0).
Note: I calculated 36% using the formula % = (1-.64)*100
Note: this does not make any sense – what does it say about strength of association?

For the Sound category main effect (p = .003): Responding round is 77% (1.77) more likely for soft sounds than for percussive sounds.
Note: What does this mean in terms of strength of association between soft/round and percussive/spiky?

For the interaction (p < .001): The difference in responding round between soft and percussive sounds is 3.06 more likely after 8 trials than after 4 trials.
Note: The authors state that the strength of association is greater at 8 trials than at 4 trials. Is this for both soft/round and percussive/spiky stimuli?

We thank Reviewer 2 for this helpful and insightful comment. We have added the odd ratio in addition to the coefficient in Table 2 and have rephrased the model reporting in the manuscript.

  1. I return to my comment in #1 about recoding the binary data. The descriptions would be much simpler. These following statements are not based on actual findings, but you can see the difference:

AT responses are more congruent after 8 trials than after 4 trials. (associations are stronger after 8 trials compared to 4 trials)

AT responses are more congruent for soft sounds than for percussive sounds (associations are stronger for soft sounds than for percussive sounds)

The difference in AT congruence (strength of association) between soft and percussive sounds is greater after 8 trials compared to 4 trials (or something like that).

We thank Reviewer 2 and agree. We have now re-framed our results in terms of congruity as the difference between effects based on sound type is not the main focus of this paper.

  1. I would like to see the results for all levels of the models: intercept only, CIM and/or AIM (intercept, main effects) and Interaction (intercept, main effects, and interaction), including a comparison of model fits. The model fits for the intercept only and final models are in the text, but I would like to see the model with main effects separately from the final model.

We thank Reviewer 2 for this comment. Parameters of all alternative models and the comparison of model fit were updated in the supplemental material (See supplemental Table 1-2).

  1. I recognize the value of the secondary analyses, but they complicate the straight-forward finds of the current study. They deserve to stand on their own. Yes, removing them would make the current study a short report, but that is ok.

We thank Reviewer 2 for this comment. We have now placed secondary analysis I in supplemental materials to streamline the focus of the current study, and integrated secondary analysis II as part of the results to tell a more coherent narrative on individual differences. We no longer refer to these results as secondary analyses.

  1. I have not commented on the discussion because it may change extensively.

The discussion has now been refocused and is now hopefully clearer.

Reviewer 3 Report

Comments and Suggestions for Authors

Review April 2025

Trials and AV and AT stimuli

This ms is about audio-tactile pairs – rounded tactile forms and spiky tactile forms with spoken words such as “baba”, “gaga” [rounded forms], “kiki”, or “titi”[spiky forms]. How many Audio-Visual trials are necessary before children of 6 to 8 show reliable Auditory-Tactile pairing? Nil and 4 are not enough, 8 produces significant pairing, as does 16. However, 8 does not prove reliable in purely Audio-Tactile experience (Figure 4, re-analyses of data from Experiment 1 of Chow et al., 2021, and though 16 looks quite reliable in Figure 4 it is deemed not to be). Conclusion: the authors “did not find significant differences in the strength of Audio-Tactile associations in 6- to 8-year-olds as a function of the amount of prior [Audio-Tactile] exposure.” That is, “simply allowing for more exposure, i.e., time-on-task or practice, does not enhance how children associate nonsense sounds and abstract shapes they explore via touch.” Usefully, “participants categorized as showing [an] expected [Auditory-Visual] association during [an] exposure phase were also more likely to be categorized as showing the expected association [Auditory-Tactile] during [a] test phase.”

Evidently the Auditory-Visual pairing is achieved more readily and it can lead to Auditory-Tactile pairing.

Theory? 1. “AV exposure enhances the ability to form visual representations of shapes…[and can] aid the processing of shapes via touch.” This may include imagery. 2. The key task is figuring out what features of shape are relevant.

Also, determination of the features of a tactile object may develop relatively slowly.  

I offer a few comments on the writing in the Introduction to this work.

The data analysis is highly professional.

The basic ideas are that over trials, participants can come to make Auditory-Tactile pairings, and though this may not occur over as many as 8 or possibly 16 trials the use of visual trials can facilitate this. These are common-sense ideas. In short, “Here we examine how much prior experience is sufficient to strengthen AT association in early development.”

No major theory is envisaged.

Abstract:

  1. “Visual experience influences AT associations.” This is vague in an Abstract and in particular one wonders if what is implied is that visual experience is actually necessary for certain classes of associations.
  2. “with more naïve visual experience.” Again, the claim is vague. Is it simply that children have had fewer years of visual experience?
  3. The children explored a shape. Then they were asked which shape matched a sound. But they only had one shape. How could they choose?
  4. Expt 1” 8 AV trials were enough to influence AT choices, 4 were not enough. But 8 AT trials in a “secondary analysis” did not change AT choices. Is there an Experiment 2? Why call an analysis a “secondary analysis?” The presentation is muddled. Could the phrase “secondary analysis be deleted? Also, another “secondary analysis” found “better” AV and “better” AT selections correlated. Please rephrase omitting the word “better”. The term is a value judgement.
  5. The final sentence states that the amount and type of experience count for AT effects. This rephases the numerical result but adds no theory.

REPORT:

  1. The first para is, understandably, a naïve version of associationism, and the authors are cautioned that associationism is largely discredited. I recommend Kukla & Walmsley (2006) on this topic. However, I will not press the point.
  2. The next sentence is circular -- things are perceived as “together” if they go together.
  3. The next sentence states that large objects are associated with low notes. This is fair.
  4. Next is the Kohler claim that “takete and maluma” (not baluma, I think) go with angular and rounded shapes respectively. Similar effects are found with “bouba and kiki.” This is sound-shape correspondence, the authors state. Similar effects are found in taste and touch. 
  5. The next sentence is about BK effects, but this is undefined. Presumably this is bouba-kiki. It is said to be due to neuronal connections strengthened in early language learning. In theory, were they actually present before “strengthening”? The effects are detected in 4-month-olds. But some “sound shape” correspondences eg to foreign language words  only emerges at 5 years of age. This section is interesting but confusing about some effects at 4 months and others only at 5. The differences in the stimuli used are not stated. The conclusion is BK effects (sound/shape)are weak at 5 years.
  6. The key question for this ms is AT – Auditory to Tactile correspondence. These are absent at 6 – 8 years and present at 9-11 years (Chow et al. 2021). Can they be “strengthened” by repetition?
  7. The ms notes that early blind participants do not show the BK effects (or only weakly) but late blind and normally sighted do. This is said to apply to objects and textures. Caution here: early blind people consider angular forms depict “hard” while curved forms depict “soft” and jerky motion is depicted by angular forms while continuous motion is by curved forms. The text goes on to state Chow found examples of visual representations of the abstract shapes strengthened judgements of correspondences. This is tautology: Visual representations just are the abstract shapes. What is meant?  Is this 2D vs 3D, as in: “As predicted, when typically-sighted children were given prior exposure with the same BK sounds and with 2-D renditions of the 3-D shapes they would later explore only via touch, only 16 trials of an AV BK task, AT associations were strengthened (Chow et al., 2021; Experiment 2).”  Were the textures sometimes depicted or were they not “represented”? The text begins with objects and textures, but the textures are not mentioned further. Are the claims only about objects? Later, raised outline drawings and “filled-in forms” (bas-reliefs, surely) are also said to be effective. The text could be made more coherent, less jumpy.
  8. Both early visual experience and time-on-task may matter, the text claims. In theory, they are alike surely.
  9. Performance is said to “improve” with practice. This is a value judgment. Perhaps only report directions and reliability, please.
  10. Curious: “blind-folded adults showed AT associations from the beginning, trial 1, but association strength diminished with repeated AT exposure.” Were they at ceiling initially? Were there sensible options, which were only adopted later? Were participants bored? Was the task too easy, and they sought interesting alternatives? Is it like repeating “horse” 20 times so it loses its apparent meaning and one only attends to its sound?
  11. At this point I will cease making detailed comments. The ms is clear but it may be working hard to support ideas that are not particularly insightful. Audio-tactile matching of unfamiliar material may be slow.  Audio-visual matches may be fast. Audio-visual practice may help participants figure out relevant shapes. The number of trials needed for practice or for figuring out is dependent on the difficulty of the task. For a given task, with given forms, there will be some number. With easier-to-distinguish and match materials the number will decrease, with harder materials it will increase. There is no fixed number of trials. An 8 or 16 result will indicate something about materials and the task and the age of participants. This can be stated at the outset.  

The experiments are done well and analyzed highly proficiently. The writing is precise.

Author Response

We thank Reviewer 3 for their careful review of our work and for the detailed and constructive comments. Below please find a point-by-point reply showing how we addressed Reviewer 3's concerns.

DETAILED COMMENTS

Abstract:

1. “Visual experience influences AT associations.” This is vague in an Abstract and in particular one wonders if what is implied is that visual experience is actually necessary for certain classes of associations.

Our ultimate goal is to determine if visual experience is necessary or sufficient, and we are finalizing studies on the type, rather than the amount, of prior exposure. For the work presented in this manuscript, we can not draw this bolder conclusion.

2. “with more naïve visual experience.” Again, the claim is vague. Is it simply that children have had fewer years of visual experience?

Yes, this is simply to state that children have less experience in the world.

3. The children explored a shape. Then they were asked which shape matched a sound. But they only had one shape. How could they choose?

We thank Reviewer 3 for highlighting this confusion. We have now revised this sentence to read “Then, children completed 16 AT trials: feeling a pair of shapes, one with a round and one with a spiky contour. Shapes were hidden from view, and children had to indicate which shape best matched a sound.”

4. Expt 1” 8 AV trials were enough to influence AT choices, 4 were not enough. But 8 AT trials in a “secondary analysis” did not change AT choices. Is there an Experiment 2? Why call an analysis a “secondary analysis?” The presentation is muddled. Could the phrase “secondary analysis be deleted? Also, another “secondary analysis” found “better” AV and “better” AT selections correlated. Please rephrase omitting the word “better”. The term is a value judgement.

We thank Reviewer 3 for this suggestion. Secondary analyses have been renamed and moved to supplemental material to streamline the focus of the current study, as requested by Reviewer 2. Thus, secondary analyses will not be mentioned in the abstract.

5. The final sentence states that the amount and type of experience count for AT effects. This rephases the numerical result but adds no theory.

At present we do not have an over-arching theory to contribute to this work. We are currently finishing up experiments examining how the type of experience matters and once we have completed this work we have a better understanding.

REPORT:

1. The first para is, understandably, a naïve version of associationism, and the authors are cautioned that associationism is largely discredited. I recommend Kukla & Walmsley (2006) on this topic. However, I will not press the point.

2. The next sentence is circular -- things are perceived as “together” if they go together.

We thank Reviewer 3 for this comment. This sentence is rephrased as “Besides spatial and temporal factors, we are also more likely to perceive select sensory information from our different senses as going together, a preferential mapping of features across the senses, known as a crossmodal correspondence (reviewed in Spence, 2011).”

3. The next sentence states that large objects are associated with low notes. This is fair.

Great. We have nothing to add here.

4. Next is the Kohler claim that “takete and maluma” (not baluma, I think) go with angular and rounded shapes respectively. Similar effects are found with “bouba and kiki.” This is sound-shape correspondence, the authors state. Similar effects are found in taste and touch. 

We thank Reviewer 3 for noting this typo. We have changed the word from “balmua” to “maluma".

4. The next sentence is about BK effects, but this is undefined. Presumably this is bouba-kiki. It is said to be due to neuronal connections strengthened in early language learning. In theory, were they actually present before “strengthening”? The effects are detected in 4-month-olds. But some “sound shape” correspondences eg to foreign language words only emerges at 5 years of age. This section is interesting but confusing about some effects at 4 months and others only at 5. The differences in the stimuli used are not stated. The conclusion is BK effects (sound/shape)are weak at 5 years.

We have now clarified that the next sentence is about BK effect. Yes, the main conclusion is that BK effects in early development. are weak at best.

5. The key question for this ms is AT – Auditory to Tactile correspondence. These are absent at 6 – 8 years and present at 9-11 years (Chow et al. 2021). Can they be “strengthened” by repetition?

We thank Reviewer 3 for this comment. Secondary analysis from Chow and colleagues (2021) did not find strengthened AT association with repetition (see Chow et al., 2021, supplemental figure 4). We highlight these effects in children at the end of Section 1.2.

6. The ms notes that early blind participants do not show the BK effects (or only weakly) but late blind and normally sighted do. This is said to apply to objects and textures. Caution here: early blind people consider angular forms depict “hard” while curved forms depict “soft” and jerky motion is depicted by angular forms while continuous motion is by curved forms. The text goes on to state Chow found examples of visual representations of the abstract shapes strengthened judgements of correspondences. This is tautology: Visual representations just are the abstract shapes. What is meant?  Is this 2D vs 3D, as in: “As predicted, when typically-sighted children were given prior exposure with the same BK sounds and with 2-D renditions of the 3-D shapes they would later explore only via touch, only 16 trials of an AV BK task, AT associations were strengthened (Chow et al., 2021; Experiment 2).”  Were the textures sometimes depicted or were they not “represented”? The text begins with objects and textures, but the textures are not mentioned further. Are the claims only about objects? Later, raised outline drawings and “filled-in forms” (bas-reliefs, surely) are also said to be effective. The text could be made more coherent, less jumpy.

We thank Reviewer 3 for highlighting this confusion. We now clarify: “one would predict that providing complementary visual exemplars of the abstract shapes might help strengthen subsequent AT associations in early childhood.” We have also removed mention of texture as there are other conditions as the reviewer notes, softness or motion trajectory, which also matter but are not the focus of this manuscript. Our 2-D shapes were not pictures of our 3-D shapes, but rather cartoon-like images. Our 3-D shapes – the 2-D shapes with added thickness, felt smooth and did not have a textured surface. We have noted this in methods where we describe our stimuli.

7. Both early visual experience and time-on-task may matter, the text claims. In theory, they are alike surely.

We would argue that these are similar but not quite the same. Visual experience, as referred to in this paper, focuses on passive exposure, while time-on-task focuses on making explicit judgements about what is observed.

8. Performance is said to “improve” with practice. This is a value judgment. Perhaps only report directions and reliability, please.

We thank Reviewer 3 for this comment. We now use the word “strengthen” instead of “improve” to avoid a value judgment.

9. Curious: “blind-folded adults showed AT associations from the beginning, trial 1, but association strength diminished with repeated AT exposure.” Were they at ceiling initially? Were there sensible options, which were only adopted later? Were participants bored? Was the task too easy, and they sought interesting alternatives? Is it like repeating “horse” 20 times so it loses its apparent meaning and one only attends to its sound?

We thank Reviewer 3 for this comment. Graven & Desebrock (2018) argued that this is due to the type of sensory experience – for early blind adults, their dominant intact sense is touch (as they are visually deprived). In this case, prior AT experience, or repeated AT testing, helps them to make better AT association. On the other hand, blindfolded sighted adults’ dominant intact sense is vision. In this case, prior AT experience, or repeated AT testing, does not work in the similar way as prior experience involving vision. We do not think boredom could explain the result in Graven & Desebrock (2018), as they are testing adult participants, and there are only 8 trials total for each participant.

Round 2

Reviewer 1 Report

Comments and Suggestions for Authors

The article has been significantly shortened and the more detailed comments I made have been addressed point by point. However, my two 'main reservations' have not been fully assuaged. My first point was that 'the present study seems to be an expansion of this earlier research, which raises the question whether it justifies an article of this length'. The shorter length does address this somewhat, but what remains still seems to repeat much of what can be found in earlier publications. Secondly, I cannot find that the authors have done anything to address the issue of p-value absolutism. I therefore still refrain from recommending publication.   

Author Response

BK QUANT REVISIONS ROUND 2

We thank the reviewers for their detailed feedback which has helped us clarify and strengthen our manuscript. Below please find our point-by-point replies to each of the reviewer’s requested revisions.

Reviewer 1

The article has been significantly shortened and the more detailed comments I made have been addressed point by point. However, my two 'main reservations' have not been fully assuaged. My first point was that 'the present study seems to be an expansion of this earlier research, which raises the question whether it justifies an article of this length'. The shorter length does address this somewhat, but what remains still seems to repeat much of what can be found in earlier publications. Secondly, I cannot find that the authors have done anything to address the issue of p-value absolutism. I therefore still refrain from recommending publication.   

We thank Reviewer #1 for these comments:

1. The current manuscript does extend upon our earlier published work (Chow et al., 2021). Some of the background information is similar, but much is unique as the research question is different. Our original research investigated how prior exposure influenced audio-tactile associations and this new manuscript looks at the question of how the amount of prior exposure matters. The original research considered two different amounts of prior exposure: no prior exposure and 16 trials of prior exposure. These two amounts of prior exposure are provided for context, showing the smallest and largest amount of exposure considered before, while the new manuscript focuses on new amounts of exposure between these extremes: 4 versus 8 trials of prior exposure. We have shortened the manuscript and elaborated on the most relevant prior work relating to our new research question. Additionally, since we have now redone all analyses to consider congruency and not vowel category, as requested by Reviewer #2, all figures and analyses now differ from the original work shown in our previous paper (Chow et al., 2021).

2. Regarding the issue with p-value absolutism, all of our statistics provide effect sizes, not just p-values. We have now emphasized the effect sizes resulting from our statistical analyses and provide a context for whether they are considered small, medium or large effect sizes, to highlight the importance of this measure.

Reviewer 2 Report

Comments and Suggestions for Authors

I thank the authors for their responsiveness to my suggestions. The major changes I suggested were to put the two vowel categories on the same scale ranging from 0 to 1 (i.e., focus on congruity) and to explain the logistic regression analyses in more detail.

The authors did explain the regression analyses sufficiently. However, the data analyses are still problematic.

As a reminder, when coding choice as round object = 1 and spiky object as 0 for both the rounded and spiky vowel categories, as in the first version of the manuscript, the meaning of strength of association differs for the two vowel categories. A score near 1.0 indicates strong association for round vowels, but a score near 0.0 indicates strong association for spiky vowels.

Instead, if choice was recoded so that 1 = expected and 0 = opposite, then for both vowel categories, a score near 1.0 indicates greater strength of association.

Although the authors did this for overall association strength (see Equation 3 below), it is unclear whether they did so for the presence/absence analysis. Further, it does not appear that the data re-coding was done for the magnitude of association analysis; in the text the authors state that the analysis was done “…to predict whether a round shape was chosen over a spiky shape” (p. 7, lines 667-668). I argue that the data should be recoded

1. Using the authors’ language, the equations for both AV and AT trials should be:

[Equation 1] Spiky vowel category AT/AV association strength = number of trials in which the spiky object was chosen, divided by number of spiky vowel category trials.

[Equation 2] Round vowel category AT/AV association strength = number of trials in which the rounded object was chosen, divided by the number of round vowel category trials.

Equation 3] Overall AT/AV association strength = number of trials in which the expected object was chosen, divided by the total number of trials.

2. For the magnitude of association analysis, the data should be recoded so that 1 = expected choice, 0 = opposite choice.

3. For the individual differences correlation analysis, the analysis is still overly complicated (i.e., comparisons with 0.5).

Instead, an association strength for AV and AT should be calculated for each participant, and these associations should be the data for the analysis. That is, each participant would have only two association strength scores, one for AV and one for AT, and all trials would be used.

This would be a much simpler correlation analysis, allowing for a direct comparison of overall AT association strength and overall AV association strength. Interpretation of the results would be simpler as well.

4. I cannot tell whether or how much the results will change, which would result in changes in the interpretation of the results. Therefore, once again, I have to refrain from evaluating discussion of the results.

However, the rest of the general discussion is fine. It sufficiently situates the current study within prior research and expands on possible explanations for the results.

Author Response

BK QUANT REVISIONS ROUND 2

We thank the reviewers for their detailed feedback which has helped us clarify and strengthen our manuscript. Below please find our point-by-point replies to each of the reviewer’s requested revisions.

Reviewer 2

I thank the authors for their responsiveness to my suggestions. The major changes I suggested were to put the two vowel categories on the same scale ranging from 0 to 1 (i.e., focus on congruity) and to explain the logistic regression analyses in more detail.

The authors did explain the regression analyses sufficiently. However, the data analyses are still problematic.

As a reminder, when coding choice as round object = 1 and spiky object as 0 for both the rounded and spiky vowel categories, as in the first version of the manuscript, the meaning of strength of association differs for the two vowel categories. A score near 1.0 indicates strong association for round vowels, but a score near 0.0 indicates strong association for spiky vowels.

Instead, if choice was recoded so that 1 = expected and 0 = opposite, then for both vowel categories, a score near 1.0 indicates greater strength of association.

Although the authors did this for overall association strength (see Equation 3 below), it is unclear whether they did so for the presence/absence analysis. Further, it does not appear that the data re-coding was done for the magnitude of association analysis; in the text the authors state that the analysis was done “…to predict whether a round shape was chosen over a spiky shape” (p. 7, lines 667-668). I argue that the data should be recoded

1. Using the authors’ language, the equations for both AV and AT trials should be:

[Equation 1] Spiky vowel category AT/AV association strength = number of trials in which the spiky object was chosen, divided by number of spiky vowel category trials.

[Equation 2] Round vowel category AT/AV association strength = number of trials in which the rounded object was chosen, divided by the number of round vowel category trials.

Equation 3] Overall AT/AV association strength = number of trials in which the expected object was chosen, divided by the total number of trials.

1. We thank Reviewer #2 for catching this problem. We have now corrected how equations are defined to make it clearer. Furthermore, we have removed equations 1 and 2, which referred to spiky and round vowel categories, as those analyses have been removed from the manuscript now that all data is represented in terms of congruent sound-shape pairs.

2. For the magnitude of association analysis, the data should be recoded so that 1 = expected choice, 0 = opposite choice.

We thank Reviewer #2 for catching this problem. We have now revised the error in the measure used for the regression analysis and report the new model statistics. The updated statistics show no major discrepancies with our original conclusions.

3. For the individual differences correlation analysis, the analysis is still overly complicated (i.e., comparisons with 0.5).

Instead, an association strength for AV and AT should be calculated for each participant, and these associations should be the data for the analysis. That is, each participant would have only two association strength scores, one for AV and one for AT, and all trials would be used.

This would be a much simpler correlation analysis, allowing for a direct comparison of overall AT association strength and overall AV association strength. Interpretation of the results would be simpler as well.

We thank Reviewer #2 for this comment. Our correlation analysis did include one measure for overall AT association and one for overall AV association per participant. We have now simplified our analysis by (1) using all AT trials to reflect the same data presented in Figure 3 in bar plots, rather than using the same number of AT trials as the number of trials in the AV exposure. The new AV-AT correlation figures and analyses include all AT test trials. (2) removing Barnard’s unconditional exact test analysis and keeping only the statistics for Kendall’s tau and Spearman’s rho to simplify the analysis. The updated statistics show no major discrepancies with our initial conclusions.

4. I cannot tell whether or how much the results will change, which would result in changes in the interpretation of the results. Therefore, once again, I have to refrain from evaluating discussion of the results.

Our conclusions remain the same.

However, the rest of the general discussion is fine. It sufficiently situates the current study within prior research and expands on possible explanations for the results.

Round 3

Reviewer 1 Report

Comments and Suggestions for Authors

I have already reviewed the manuscript two times and recommended a rejection. I feel it puts undue pressure on me to ask me to review the manuscript for the third time. Two times 'reject' means 'reject'. I am perfectly happy if the editor is willing to select another referee.

Author Response

No further edits were suggested by Reviewer 2. Thank you.

Reviewer 2 Report

Comments and Suggestions for Authors

I thank the authors again for their attention to my comments. The revisions have addressed all my concerns.

Now I can comment on the general discussion, and I have three issues to be addressed.

1. The first involves section 3.2 What is being learned during AV exposure?

I am not sure I understand the paragraphs describing the two possibilities. At the very least, the two possibilities need to be clarified.

If I understand correctly, both of these possibilities are based on classical conditioning learning principles—an association is created between a visual object and a sound (or vice versa). The more trials, the stronger the association.

The following is more for my benefit than the authors, but I hope I add more focus to the possible ways of learning the association. I do not expect the authors to incorporate this information—just make the two possibilities (and their implications for learning intervention) clearer.

In either possibility, the visual object could be represented in memory as a visual mental image or a non-visual list of object characteristics. We do not know much about how sounds are remembered.

One possibility was that the participant focused on the sound as the stem of the association, creating a sound—visual object association.

Sound A – Visual Object A1 and Visual Object A2 (or vice versa)

Sound B – Visual Object B1 and Visual Object B2 (or vice versa)

At test, the participant focuses first on the sound, recalls the visual object associated with it, matches the characteristics of the remembered visual object with the characteristics of the felt objects, and chooses the tactile object with the same characteristics.

The second possibility was that the participant focused on the visual objects as the stem of the association, creating a visual object – sound association.

Visual Object A1 and Visual Object A2 – Sound A

Visual Object B1 and Visual Object B2 –  Sound B

At test, the participant feels the shapes, matches its characteristics with the remembered visual shapes, recalls the sound associated with each shape, matches the heard sound with the remembered sounds to choose the tactile object associated with the matched sound.

So, what would determine the individual differences in the strength of associations?

  1. Differences in paired association learning in general.
  2. Differences in visual short-term or working memory (visual representations)
  3. Differences in verbal short-term or working memory (non-visual representations)
  4. Differences in short-term or working memory for sounds

2. The second issue is about section 3.3. The Development of Haptic-to-Visual Transfer.

I am not sure why this section is included because the current study focused on visual-to-haptic transfer. The section should be “Development of Visual-to-Haptic Transfer.”

At the very least, explain why the focus is on haptic-to-visual rather than visual-to-haptic.

3. The last issue involves terminology in 3.3 when describing the image-mediation model and haptic-to-visual transfer.

For clarification, use “visual depiction of an object” when referring to a physically-present visual object and “visual mental image of an object” when referring to the representation of the physical object in memory.

Author Response

I thank the authors again for their attention to my comments. The revisions have addressed all my concerns.

Now I can comment on the general discussion, and I have three issues to be addressed.

  1. The first involves section 3.2 What is being learned during AV exposure?

I am not sure I understand the paragraphs describing the two possibilities. At the very least, the two possibilities need to be clarified.

If I understand correctly, both of these possibilities are based on classical conditioning learning principles—an association is created between a visual object and a sound (or vice versa). The more trials, the stronger the association.

The following is more for my benefit than the authors, but I hope I add more focus to the possible ways of learning the association. I do not expect the authors to incorporate this information—just make the two possibilities (and their implications for learning intervention) clearer.

In either possibility, the visual object could be represented in memory as a visual mental image or a non-visual list of object characteristics. We do not know much about how sounds are remembered.

One possibility was that the participant focused on the sound as the stem of the association, creating a sound—visual object association.

Sound A – Visual Object A1 and Visual Object A2 (or vice versa)

Sound B – Visual Object B1 and Visual Object B2 (or vice versa)

At test, the participant focuses first on the sound, recalls the visual object associated with it, matches the characteristics of the remembered visual object with the characteristics of the felt objects, and chooses the tactile object with the same characteristics.

The second possibility was that the participant focused on the visual objects as the stem of the association, creating a visual object – sound association.

Visual Object A1 and Visual Object A2 – Sound A

Visual Object B1 and Visual Object B2 – Sound B

At test, the participant feels the shapes, matches its characteristics with the remembered visual shapes, recalls the sound associated with each shape, matches the heard sound with the remembered sounds to choose the tactile object associated with the matched sound.

So, what would determine the individual differences in the strength of associations?

  1. Differences in paired association learning in general.
  2. Differences in visual short-term or working memory (visual representations)
  3. Differences in verbal short-term or working memory (non-visual representations)
  4. Differences in short-term or working memory for sounds
  5.  

We thank Reviewer 2 for providing a detailed elaboration on what could account for individual differences and highlighting the confusion in the two possibilities we presented in section 3.2. We have now clarified the two possibilities we presented and mention that various mechanisms could underly individual differences. We do not include discussion of the possibility that the visual object could be represented in memory as a non-visual list of object characteristics as it seems too broad for the current work.

The way we conceptualize the first possibility does not involve classical conditioning. Rather, the contrast between shape contours is highlighted by visual experience which should then strengthen the reliability of visual information / visual mental imagery vividness. Thus, the visual part of the prior experience could then allow the tactile shape to be better represented as a visual mental image, and allow it to be more easily associated with a corresponding sound. Studies in congenitally blind individuals who show impaired AT associations would suggest that this visual imagery part is important in audio-tactile associations. The second possibility focuses on the role of the nonsense sounds, as the reviewer highlights in the first possibility they present.

  1. The second issue is about section 3.3. The Development of Haptic-to-Visual Transfer. I am not sure why this section is included because the current study focused on visual-to-haptic transfer. The section should be “Development of Visual-to-Haptic Transfer.”

At the very least, explain why the focus is on haptic-to-visual rather than visual-to-haptic.

We thank Reviewer 2 for this comment. Our rationale for discussing haptic-to-visual transfer was to emphasize possible benefits for processing shape features via touch during the AT task. According to the image-mediation model of haptic processing, haptic inputs are typically translated into a visual mental image before further processing. Here we suggest that prior AV exposure, primarily seeing the shape, may facilitate the transfer of haptic inputs (i.e. shape features acquired via touch during the AT task) into a visual image which then helps in associating the touched shape with a nonsense sound. Here visual would help haptic-to-visual which then helps audio-haptic associations. We have now revised section 3.3 to better highlight our underlying rational and have changed the header to “The Development of Haptic Processing” as the focus is broader.

  1. The last issue involves terminology in 3.3 when describing the image-mediation model and haptic-to-visual transfer. For clarification, use “visual depiction of an object” when referring to a physically-present visual object and “visual mental image of an object” when referring to the representation of the physical object in memory.

We thank Reviewer 2 for highlighting this important issue. We have now made clearer the distinctions between “visual depiction of an object” and “visual mental image of an object” when discussing the image-mediation model and haptic-to-visual transfer.
